# Development of Synthetic mRNAs Encoding Split Cytotoxic Proteins for Selective Cell Elimination Based on Specific Protein Detection

**DOI:** 10.3390/pharmaceutics15010213

**Published:** 2023-01-07

**Authors:** Kendall Free, Hideyuki Nakanishi, Keiji Itaka

**Affiliations:** Department of Biofunction Research, Institute of Biomaterials and Bioengineering, Tokyo Medical and Dental University (TMDU), Tokyo 101-0062, Japan

**Keywords:** mRNA, intein, nanobody, selective cell elimination

## Abstract

For the selective elimination of deleterious cells (e.g., cancer cells and virus-infected cells), the use of a cytotoxic gene is a promising approach. DNA-based systems have achieved selective cell elimination but risk insertional mutagenesis. Here, we developed a synthetic mRNA-based system to selectively eliminate cells expressing a specific target protein. The synthetic mRNAs used in the system are designed to express an engineered protein pair that are based on a cytotoxic protein, Barnase. Each engineered protein is composed of an N- or C-terminal fragment of Barnase, a target protein binding domain, and an intein that aids in reconstituting full-length Barnase from the two fragments. When the mRNAs are transfected to cells expressing the target protein, both N- and C-terminal Barnase fragments bind to the target protein, causing the intein to excise itself and reconstitute cytotoxic full-length Barnase. In contrast, when the target protein is not present, the reconstitution of full-length Barnase is not induced. Four candidate constructs containing split Barnase were evaluated for the ability to selectively eliminate target protein–expressing cells. One of the candidate sets demonstrated highly selective cell death. This system will be a useful therapeutic tool to selectively eliminate deleterious cells.

## 1. Introduction

Cytotoxic genes can be used to eliminate deleterious cells, such as cancer or virus-infected cells. However, the expression of the cytotoxic gene in healthy cells can cause undesirable off-target cell death. Therefore, selective cell elimination systems are needed to target and eliminate only deleterious cells. In previous studies, cell-selective transcriptional regulatory sequences have been used to make selective cell elimination systems. Despite the success of such DNA-based systems in achieving selective cell elimination [1], the risk of genomic integration limits their applicability. Alternatively, the development of mRNA-based systems improves this safety concern [2,3]. Some mRNA-based systems achieve selective elimination after recognizing microRNA (miRNA) specific to a target cell type [4,5,6]. Although these miRNA-detecting systems successfully worked in certain situations, detecting intracellular proteins seems to be a more practical approach as there are abundant intracellular proteins specifically expressed in deleterious cells (e.g., cancer antigens and viral proteins).

Therefore, we developed a synthetic mRNA-based intracellular protein-detecting system to achieve selective cell elimination. This system is composed of two mRNAs encoding fusion proteins, which consist of N- or C-terminal fragments of Barnase (Bn), a cytotoxic protein [7]; caged inteins that can excise themselves from the host protein and join the flanking regions through a peptide bond once in proximity [8,9]; and nanobodies, which are antigen-binding domains of camelid-derived single-chain antibodies [10]. In this system, they force the N- and C-caged inteins into proximity by tethering them to a target protein. When the mRNAs encoding the engineered proteins are transfected into cells expressing the target protein, a binding event occurs, causing the intein to excise itself and reconstitute full-length Bn. The selective reconstitution of Bn enables target protein–responsive cytotoxicity (Figure 1).

## 2. Materials and Methods

### 2.1. pDNA Construction

The human codon-optimized Barstar-Barnase DNA was synthesized by GeneArt High-Q strings (ThermoFisher Scientific K.K., Tokyo, Japan). PrimeSTAR Max DNA Polymerase (Takara Bio Inc., Shiga, Japan) was used for the polymerase chain reaction (PCR), to prepare the inserts by using human codon-optimized Barstar-Barnase as template DNA. Primers for each of the split Bn were designed using Primer3Plus [11]. Vectors digested by restriction enzymes and inserts were purified using the Monarch PCR & DNA Cleanup Kit (New England BioLabs Japan Inc., Tokyo, Japan). The cloning reaction was performed using the In-Fusion HD Cloning Kit (Takara Bio Inc.). Further, pDNAs were amplified in *E. coli* strain HST08 and extracted using the Monarch Plasmid Miniprep Kit (New England BioLabs Japan Inc.).

### 2.2. Preparation of mRNAs by In Vitro Transcription

Template DNAs were prepared by PCR using PrimeSTAR Max DNA Polymerase (Takara Bio) and the primers listed in Appendix A, and the Monarch PCR & DNA Cleanup Kit (New England BioLabs Japan Inc.) was used to purify the PCR products. Template DNA sequences used in this study are shown in Appendix A. The MEGAscript T7 Transcription Kit (Thermo Fisher Scientific K.K.) was used to transcribe mRNAs from the template DNAs. Additionally, 6 mM GTP, 6 mM CTP, and 6 mM ATP were used from the MEGAscript T7 Transcription Kit (Thermo Fisher Scientific) with 6 mM N1-methyl-pseudoUTP (TriLink Biotechnologies, San Diego, CA, USA) and 4.8 mM CleanCap Reagent AG (3′ OMe) (TriLink Biotechnologies). After transcription, template DNAs were digested using TURBO Dnase (Thermo Fisher Scientific), and transcribed mRNAs were purified using Agencourt RNAClean XP (Agencourt Bioscience Corporation, Beverly, MA, USA). Next, the mRNAs were dephosphorylated using Quick CIP (New England BioLabs Japan) and purified using the RNeasy Mini Kit (Qiagen K.K., Tokyo, Japan). The mRNAs were quantified using NanoDrop One (ThermoFisher Scientific K.K.), and their sizes were confirmed using the Agilent RNA 6000 Nano Assay and the Agilent 2100 Bioanalyzer (Agilent Technologies Japan Ltd., Tokyo, Japan).

### 2.3. Cell Culture

HeLa cells were cultured in Dulbecco’s modified Eagle’s medium (4500 mg/L glucose, L-glutamine, sodium pyruvate and sodium bicarbonate) (Sigma Aldrich Japan K.K., Shinagawa, Tokyo) containing 10% fetal bovine serum (Biosera, Tokyo, Japan) and 1% penicillin-streptomycin (Sigma Aldrich Japan K.K.).

### 2.4. The mRNA Transfection and Cell Viability Assay

HeLa cells were seeded at a density of 1 × 10^4^ cells/well in a 96 well plate. Approximately 24 h later, cells were transfected using 0.2 µL/well of Lipofectamine MessengerMAX (Thermo Fisher Scientific K.K.). Cell viability was measured 24 or 48 h after transfection using the Cell Counting Kit-8 Assay (Dojindo Laboratories, Kumamoto, Japan).

## 3. Results

### 3.1. Construction of Split Bn Genes

To successfully reconstitute Bn from the N- and C-terminal split parts, different candidate split Bn genes were constructed. By following previous studies using split versions of Bn [12], one split Bn gene candidate was split at the 36th amino acid (Bn-36). Next, Split Protein rEassembly by Ligand or Light (SPELL) [13] predicted a suitable split site at the 21st amino acid (Bn-21). The remaining split sites were selected because of their positions in the middle of long, unstructured regions within the Bn protein (Bn-65 and Bn-81), which would theoretically allow for optimal full-length protein reconstitution [14]. *Escherichia coli* dihydrofolate reductase (eDHFR) was used as a target protein for this system because of the strong performance of eDHFR-binding nanobodies in previous studies [15,16]. In order to bind the N-terminal Bn fragments to eDHFR, the fragments were fused with an eDHFR α epitope-targeting nanobody Nb113, and the C-terminal Bn fragments were fused with eDHFR β epitope-targeting nanobody CA1698. Further, the caged *Nostoc punctiforme* (Npu) DnaE intein [8,9,17] was inserted between the split Bn and nanobody portions of the engineered mRNAs. The caged Npu DnaE intein was used to mediate fast protein trans-splicing, which reconstitutes full-length Bn from the N- and C-terminal split fragments when the N- and C-caged inteins are in proximity. This split intein requires the first amino acid of the C-extein to be cysteine [18], so we inserted a cysteine codon into the splice junction of the C-extein. Then, we transfected these split Bn mRNA pairs to evaluate their selective cell elimination capability. Because we previously confirmed the high mRNA transfection efficiency in HeLa cells [16], we selected HeLa cells to demonstrate this systems’ proof of concept.

### 3.2. Selective Cytotoxicity after Cotransfecting Split Bn Fragments

Each of the mRNAs encoding split protein fragments were transfected into HeLa cells, and the cell viability was measured 24 h after transfection. All split Bn mRNAs reduced the cell viability compared with the untransfected condition (Figure 2A). However, there were differences in the performance between each of the split Bn constructs. The fold change between the on-target condition, which expressed the target protein, and the off-target condition, which lacked the target protein, were calculated. Bn-81 showed a 4.8-fold change between the on- versus off-target conditions, the highest of any split Bn set. This indicated highly selective cell elimination in response to the dimerizing target protein, eDHFR. In contrast, Bn-65 had high cytotoxicity for both the on- and off-target conditions and eliminated approximately 95% of cells. Bn-21 showed the lowest overall cytotoxicity.

Although Bn-81 showed the highest fold change in cytotoxicity between the target protein-positive and -negative cells, more than 80% of cells were eliminated even when there was no target protein. In particular, 48 h after transfection, most of the cells died, regardless of target protein expression (Figure 2B). As such off-target cell death is undesirable for therapeutic applications, we investigated how to prevent it.

### 3.3. Prevention of Off-Target Cell Death by Cotransfection of Barstar mRNA

Despite Bn-81 showing the highest fold change between the on- and off-target conditions, only approximately 5% of cells in the off-target condition lacking the target protein were not eliminated (Figure 2B). We speculated that a small portion of split Bn was reconstituted into full-length Bn by the target protein-independent protein splicing and that the off-target cytotoxicity results from such reconstituted full-length Bn. To reduce this off-target cytotoxicity, we cotransfected Bn-81 with Barstar, a protein sterically inhibits Bn by forming a one-to-one, noncovalently bound complex (Figure 3A) [19,20,21].

The smallest amount of Barstar mRNA necessary to reduce off-target cytotoxicity was evaluated. Cotransfection of 3, 2, or 1 ng of Barstar mRNA with Bn-81 resulted in nearly 100% viability of target protein-lacking cells, while high cytotoxicity was still observed in the target protein-expressing cells, especially when 1 ng of Barstar mRNA was used (Figure 3B). These results demonstrate the superiority of Bn-81 in its ability to selectively eliminate cells that are expressing the target protein while it maintains limited off-target cytotoxicity.

### 3.4. Relationship between Amount of Target Protein and Cell Elimination Efficiency

To investigate the sensitivity of this system for detecting different amounts of intracellular protein, Bn-81 was cotransfected with 3, 2, 1, or 0.5 ng of eDHFR mRNA. There was a dose-dependent relationship between the amount of eDHFR mRNA transfected and cell viability. When 0 ng of eDHFR mRNA was used, the cell viability was similar to the transfection reagent-only condition (Figure 4). Only 2 ng of eDHFR mRNA was necessary for approximately 50% of the cells to be eliminated, and 3 ng of eDHFR mRNA eliminated approximately 80% of the cells. Therefore, this system demonstrates high sensitivity and that it selectively eliminated cells in a target protein-dependent manner.

### 3.5. Generality of Protein-Responsive Selective Cell Elimination

Lastly, we used a different intracellular protein, EGFP, to check the generality of this system. The regions encoding anti-eDHFR nanobodies were changed to Lag16 [22] and the GFP enhancer nanobody [23] (Figure 5A). We chose this nanobody combination because these nanobodies bind to different epitopes of EGFP and therefore do not inhibit each other’s binding [16,24]. As in the case of eDHFR-responsive Bn-81, selective cell elimination occurred only in the target conditions where EGFP was present (Figure 5B).

## 4. Discussion

In this study, we developed a synthetic mRNA-based system to selectively eliminate target protein-expressing cells. Here, mRNA encoding Bn, a cytotoxic protein, was split into two parts and fused with genes encoding a nanobody and caged intein for intracellular protein recognition and full-length protein reconstitution from the split parts. Four candidate split Bn genes were designed, and Bn-81 showed a 4.8-fold change between target protein-expressing cells and target protein-lacking cells, which was the highest of the candidate genes (Figure 2A). However, there was undesirable cytotoxicity even in the target protein-lacking cells, which were the off-target condition. Because of this off-target cytotoxicity, the cell viability in the off-target condition was only approximately 11% and 5% 1 and 2 days after transfection, respectively (Figure 2). Therefore, to reduce this off-target cytotoxicity, we cotransfected Barstar, an inhibitor of Bn (Figure 3A), with Bn-81 and achieved nearly 100% cell viability (Figure 3B).

Compared with the recently reported mRNA-based selective cancer cell elimination systems, our split cytotoxic protein system is relatively simple [25,26]. Nevertheless, the need to cotransfect mRNA encoding an inhibitor protein, Barstar, to reduce off-target cytotoxicity complicates this system. As Bn is a highly cytotoxic protein, even inefficient target protein-independent spontaneous protein splicing may be sufficient to induce undesirable off-target cytotoxicity. Using other cytotoxic proteins, such as Bim [4] or Bax [5], may simplify this system because these proteins might not cause off-target cytotoxicity without an inhibitor. Further optimizing the used caged inteins may likewise decrease the off-target cytotoxicity by decreasing the target protein–independent protein splicing.

There are several limitations to this system. First, as Bn can eliminate only the cells it is transfected into, it is not capable of inducing bystander effects. Therefore, eliminating all deleterious cells in vivo using only this system is challenging, especially in the case of solid tumors with hypoxic regions, because transfecting all target cells with mRNAs is difficult. Developing the split version of prodrug-converting enzymes (e.g., herpes simplex virus thymidine kinase [27,28]) instead of Bn may overcome this limitation, but their bystander effects can kill nontarget cells. Thus, there is a trade-off between high cell selectivity and the bystander effect. Determining which of these considerations should be prioritized depends on the purpose. Second, it is difficult to target proteins inside some membranous organelles, such as lysosomes or peroxisomes. The localization of split Bn to such organelles may be possible by fusing signal peptides. However, the reconstitution of full-length Bn in such organelles seems to not efficiently induce cell death, because Bn induces cell death by nonspecific RNA degradation, and RNAs are mainly localized in the cytoplasm and nucleus.

The cytotoxicity that is due to reconstituted Bn is based on the presence of an intracellular target protein, and the detection of the target protein depends solely on the nanobody. Therefore, by changing the nanobody used, various cell-specific intracellular proteins can be targeted. This enables the selective elimination of various deleterious cells expressing specific proteins (e.g., cancer antigens or viral proteins). At present, there is a limited number of nanobodies are available [29,30], which restricts this work to a model system. However, new nanobodies that target endogenous cell-specific proteins can be obtained by immunizing camelids. Alternatively, other customizable protein-targeting modules such as DARPin [31,32], affibody [33], and anticalin [34] can be used to bring the inteins into proximity and reconstitute the full-length cytotoxic protein. A noncustomizable protein, such as a naturally occurring protein and binding pair, could also potentially be used [25]. The versatility of this system lends itself to various clinical applications. For example, using a protein-targeting module that detects a cancer antigen may allow for this system to be used for cancer therapy [25,26,35]. The ability of this system for such in vivo selective cell elimination could be evaluated by using a tumor xenograft model. Additionally, it could be used to aid regenerative medicine by eliminating undifferentiated cells in mixed-cell populations before transplanting these cells [4,36]. Thus, this study shows a promising proof of concept for using intracellular protein detection as a basis for an mRNA-based selective cell elimination system.

## Figures and Tables

**Figure 1 pharmaceutics-15-00213-f001:**
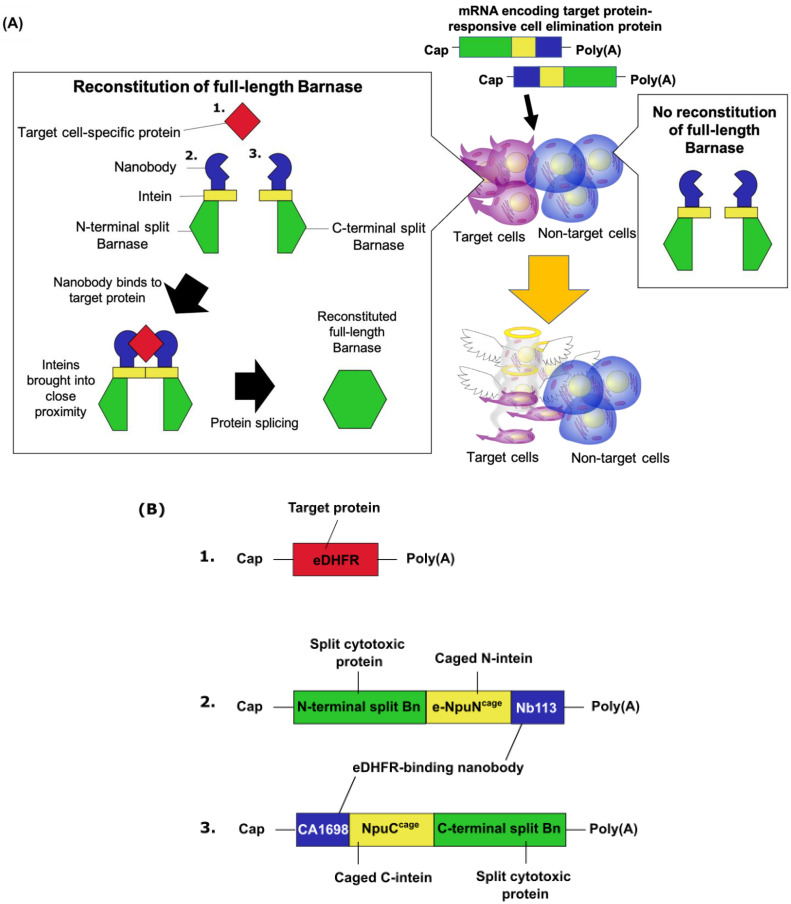
Mechanism of selective cell elimination using mRNA encoding split Bn. (**A**) Schematic diagram of selective cytotoxicity after transfecting the mRNA encoding target protein-responsive cell elimination protein. (**B**) Schematic diagram of mRNAs used in the target protein-responsive cell elimination system. All mRNAs were capped with CleanCap AG (3′ OMe).

**Figure 2 pharmaceutics-15-00213-f002:**
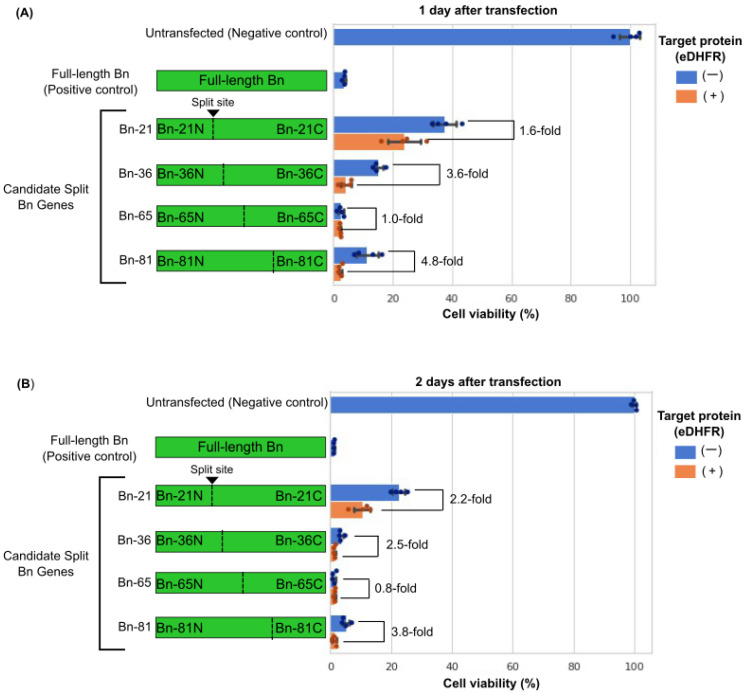
Cell viability assay results comparing the selective cytotoxicity of each candidate split Bn set. (**A**) Cell viability 1 day after transfecting candidate split Bn sets. HeLa cells were cotransfected with each of the N- and C-terminal split Bn sets (45 ng/well of each) and eDHFR mRNA (10 ng/well). Full-length Bn (90 ng/well) was used as a positive control. (**B**) Cell viability 2 days after transfecting candidate split Bn sets.

**Figure 3 pharmaceutics-15-00213-f003:**
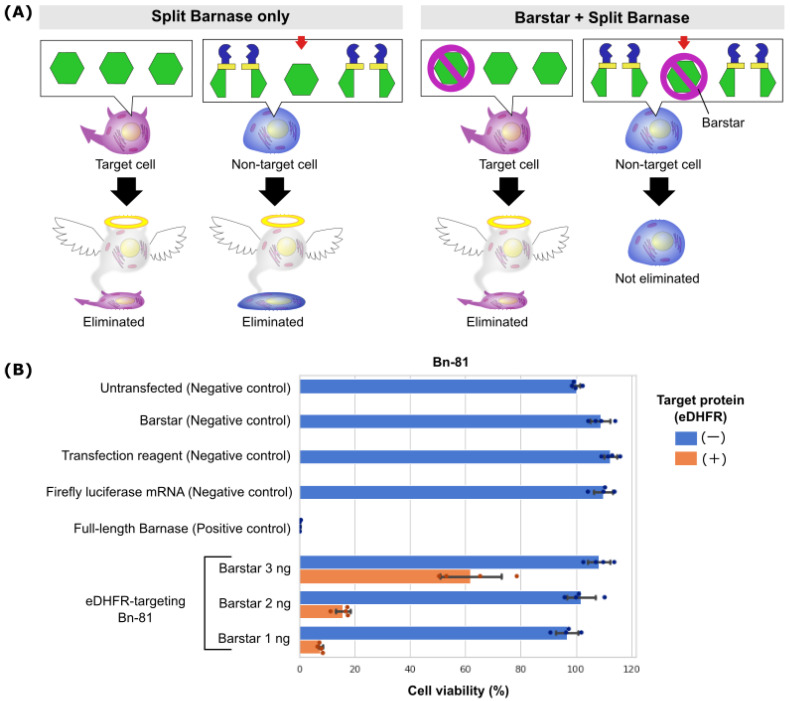
Reduction of off-target cytotoxicity by cotransfection of Barstar. (**A**) Schematic diagram showing reduced off-target cytotoxicity by cotransfection of Barstar. A small portion of split Bn can be reconstituted into full-length Bn even in nontarget cells (indicated with a red arrow), which causes undesirable off-target cytotoxicity. The cotransfection of a small amount of Barstar can sufficiently inhibit the full-length Bn in nontarget cells. In contrast, owing to high protein splicing efficiency in target cells, such a small amount of Barstar is insufficient to inhibit on-target cytotoxicity. (**B**) Cell viability assay results from cells transfected with Bn-81 and Barstar mRNA. Cell viability was measured 2 days after cotransfecting HeLa cells with Bn-81 (45 ng/well of both N- and C-terminal split Bn-81), eDHFR (10 ng/well) and Barstar mRNA (3, 2, or 1 ng/well). Firefly luciferase mRNA (90 ng/well), Barstar mRNA (90 ng/well), and the transfection reagent without mRNA were used as negative controls. Full-length Bn (90 ng/well) was used as a positive control.

**Figure 4 pharmaceutics-15-00213-f004:**
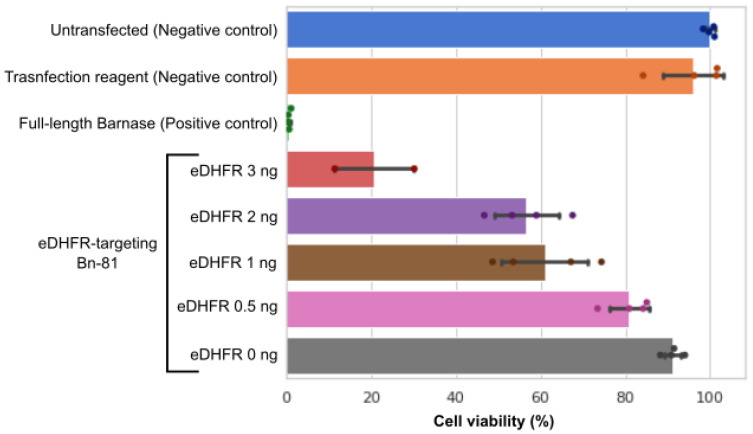
Relationship between the amount of transfected eDHFR mRNA and reconstitution of Bn protein, as measured by cell viability. Cell viability was measured 2 days after cotransfecting Bn-81 mRNA (45 ng/well of both N- and C-terminal Bn-81), Barstar mRNA (1 ng/well), and eDHFR mRNA (3, 2, 1, 0.5, or 0 ng/well). The transfection reagent without mRNA was used as a negative control, and full-length Bn mRNA (90 ng/well) was used as a positive control.

**Figure 5 pharmaceutics-15-00213-f005:**
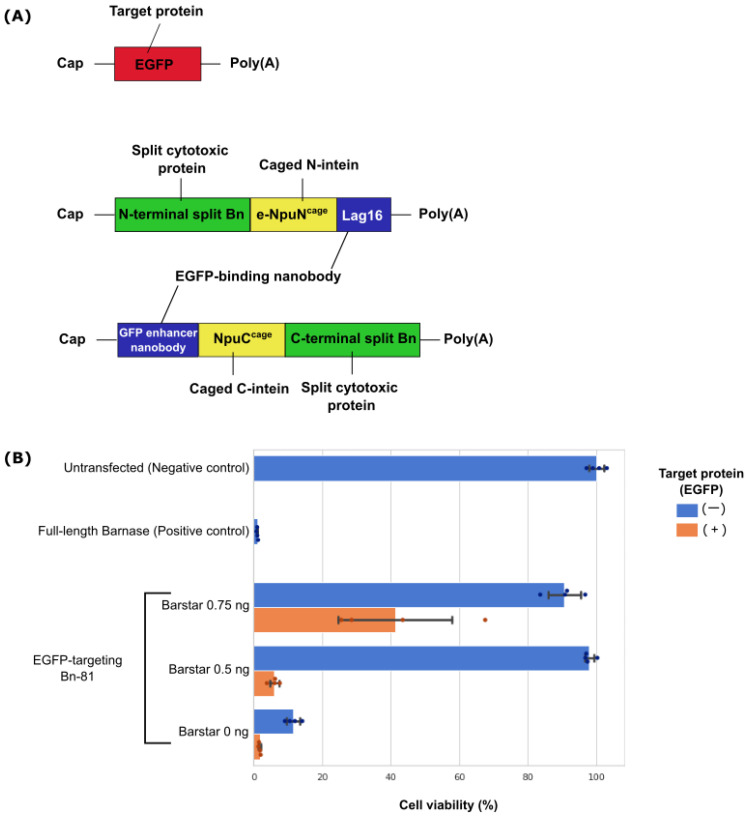
EGFP-targeting Bn-81 selectively eliminates target cells. (**A**) Schematic diagram of EGFP-targeting mRNAs used. All mRNAs were capped using CleanCap AG (3′ OMe). (**B**) Cell viability 2 days after cotransfecting HeLa cells with EGFP-targeting Bn-81 (45 ng/well of both N- and C-terminal split EGFP-targeting Bn-81), EGFP (10 ng/well), and Barstar mRNA (0.75, 0.5, or 0 ng/well). Full-length Bn (90 ng/well) was used as a positive control.

## Data Availability

The raw data are available from the lead contact upon request.

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
