# Peer review of "Development of Synthetic mRNAs Encoding Split Cytotoxic Proteins for Selective Cell Elimination Based on Specific Protein Detection"

_pharmaceutics, 2023, doi:10.3390/pharmaceutics15010213_

Round 1

Reviewer 1 Report

In this manuscript, Free et al. developed a synthetic mRNA-based system to eliminate cells expressing a specific target protein, which could be used for targeted cancer cell death. Therefore, two mRNAs encoding fusion proteins for N- or C-terminal fragment of barnese (a cytotoxic protein), caged inteins, and target binding nanobodies were designed. In the presence of the target protein, full-length barnase should be reconstituted; in the absence of the target protein, barnase should not be reconstituted. In the presence of the target protein, full-length barnase should be reconstituted; in the absence of the target protein, barnase should not be reconstituted. This idea is excellent for enabling target cell specific protein production. This can be used to eliminate off-target effects of mRNAs encoding cell toxic proteins. The effect was evaluated only by detecting the cytotoxicity. However, an off-target effect was still detected even though the target protein was not present. The production of barnase protein in cells was not detected and should be demonstrated. In cells with target protein, barnase should be detected as a full-length protein, and in cells without target protein, fragments of barnase should be present. Also, protein expression of eDHFR and barstar after mRNA transfections is not shown. These important experiments are missing. In addition, it is also not clear how barstar eliminates the off-target effect of barnase. It would be helpful, if the autors explained the mechanism behind this. Furthermore, mRNAs without intein and nanobody parts should be used as controls.

Reviewer 2 Report

Free et al. developed a synthetic mRNA based intracellular protein detecting system to eliminate deleterious cells. They used Bn cytotoxic protein as a target cell specific model to explore the mRNA based selective cell elimination. When mRNA is transfected into the cells expressing target protein leads to intein excised and reconstitute full length Bn from two mRNA encoding fusion proteins. Thus the split Bn mRNA is efficient in cell elimination capability. It will be interesting to explore the packaging of mRNA into extracellular vesicles to inject into mice to test its direct clinical relevance (this is my suggestion, not required to addressed in this article). I am particularly motivating the authors for discussion part to explain a little more with related contents that will make more sense to the readers of Pharmaceutics.

 I have few comments:

Why authors used only HeLa cells? Just for transfection efficiency?

Authours can validate their finding through fluorescence microscopy?

This study don’t clearly show the efficiency of this methos?

How this method is applicable in-vivo?

Authors should briefly discuss limitation of this study.

Round 2

Reviewer 1 Report

Thank you for the revision and clarifications.